# Characteristics of Far-Infrared Ray Emitted from Functional Loess Bio-Balls and Its Effect on Improving Blood Flow

**DOI:** 10.3390/bioengineering11040380

**Published:** 2024-04-15

**Authors:** Yeon Jin Choi, Woo Cheol Choi, Gye Rok Jeon, Jae Ho Kim, Min Seok Kim, Jae Hyung Kim

**Affiliations:** 1R&D Center, Hanwool Bio, Yangsan 50561, Republic of Korea; hbio1004@naver.com (Y.J.C.); lih1769@naver.com (W.C.C.); 2R&D Center, eXsolit, Yangsan 50611, Republic of Korea; grjeon@pusan.ac.kr (G.R.J.); jhkim@pusan.ac.kr (J.H.K.); 3Monash Health, Melbourne, VIC 3800, Australia

**Keywords:** loess particles, functional loess bio-ball, far-infrared ray (FIR) therapy, blood improvement, blood circulation

## Abstract

XRD diffraction and IR absorption were investigated for raw loess powder and heat-treated loess powder. Raw loess retains its useful minerals, but loses their beneficial properties when calcined at 850 °C and 1050 °C. To utilize the useful minerals, loess balls were made using a low-temperature wet-drying method. The radiant energy and transmittance were measured for the loess balls. Far-infrared ray (FIR) emitted from loess bio-balls is selectively absorbed as higher vibrational energy by water molecules. FIR can raise the body’s core temperature, thereby improving blood flow through the body’s thermoregulatory mechanism. In an exploratory study with 40 participants, when the set temperature of the loess ball mat was increased from 25 °C to 50 °C, blood flow increased by 39.01%, from 37.48 mL/min to 52.11 mL/min, in the left middle finger; in addition, it increased by 39.62%, from 37.15 mL/min to 51.87 mL/min, in the right middle finger. The FIR emitted from loess balls can be widely applied, in various forms, to diseases related to blood flow, such as cold hands and feet, diabetic foot, muscle pain, and menstrual pain.

## 1. Introduction

Loess is a naturally occurring sediment containing a porous mixture of various minerals like silicic acid, iron oxide, and anhydride. It is generally composed of quartz (60–70%), feldspar and mica (10–20%), carbonate minerals (5%), and silt (2–5%), and emits a large amount of far-infrared ray (FIR) in its natural state [1]. Loess contains useful microbial enzymes such as catalase, diphenol oxidase, saccharase, and protease [2]. Soil enzymes are natural mediators and catalysts for many important soil processes and perform a wide range of functions within living organisms, such as metabolic processes, cell signaling and regulation, cell division, and cell death [3]. Loess has been widely used for many centuries in traditional Korea for buildings, tools, kitchen utensils, and even as a medical therapy in the royal court [1,4]. In recent years, the use of this versatile sediment has expanded, and loess can now be found in clothing, as a natural pigment; health foods; cosmetics; eco-friendly building materials; and bio-pesticides that reduce red algae in the sea [5,6]. However, the literature on the use of loess is inadequately described in historical records, and scientific research on the clinical efficacy of loess is still in the early stages [7].

FIR (λ = 4–1000 μm, 12.4 meV–1.7 eV) is a subdivision of the electromagnetic spectrum that has been investigated for its biological effects [8]. FIR can transfer heat to deep areas of the human body by radiation because it has a higher energy efficiency than surface thermal conduction [9]. It can penetrate up to 4–5 cm from the epidermis, where it is absorbed into tissues and generates a thermal effect by increasing the vibrational motion of water molecules [8]. Studies have shown that FIR with a wavelength of 4–16 μm non-specifically raises body-tissue temperature and improves body-fluid motility by reducing the size of water clusters in vivo [10]. Since FIR is in the same frequency range as the natural frequency of water molecules in biological tissues, its effect can be further amplified in human tissues through the resonance of waves [11]. Because FIR is emitted from various materials around us, extensive research has been conducted on luminescent materials and their effects [12]. Recently, the use of FIR using loess has expanded not only to housing materials and household items, but also to bedding, auxiliary equipment, and alternative medical devices [13,14].

The chemical components of raw loess and how these components change during low- and high-heat treatment were investigated in this study. Experimental results confirm that loess retains its original minerals at low temperatures but undergoes significant chemical change and loses many of its beneficial properties at high temperatures. To utilize the useful components found in natural loess, functional loess balls were manufactured using a low-temperature wet-drying method. In order to apply loess balls for health promotion purposes, the radiant intensity and emissivity of FIR emitted from the loess balls were obtained in the wavelengths range of 5 to 20 μm. In addition, the transmitted energy and transmittance of FIR through five different types of materials were measured to evaluate the suitability of materials needed to manufacture health promotion products using loess bio-balls. Since the emission frequency (or wavelength) of FIR emitted from the loess ball is similar to the absorption frequency (or wavelength) of water molecules in the body, the water molecules selectively absorb this FIR and effectively raise the core temperature of the body, thereby increasing blood flow and epidermal temperature. Other studies have reported changes in skin surface temperature and blood flow before and after applying heat [15] or FIR [16] to the skin surface, but these changes are mainly caused by conductive heat applied to the skin surface. It is analyzed as an increase in blood flow or temperature that occurs locally/temporarily in the epidermis. In an exploratory experiment conducted on 40 participants, blood flow and epidermal temperature in the middle fingers of both hands were measured using a laser Doppler flowmeter and a non-contact infrared thermometer while gradually increasing the set temperature of the loess bio-ball mat from 25 °C to 50 °C. Experimental results were analyzed in terms of the temperature-regulation mechanism of the hypothalamus.

## 2. Materials and Methods

### 2.1. X-ray Diffraction (XRD)

The crystalline state of the loess powder was investigated using an X-ray diffractometer (XRD, Rigaku miniflex 600, Tokyo, Japan). The chemical substances and composition ratios in the loess powder were also obtained in this measurement. This experiment was measured at the JS Tech laboratory, Tokyo, Japan. X-ray diffraction experiments to investigate the phase change of raw and heat-treated loess powder was performed using a Rikaku miniflex 600 at a Makino laboratory, Tokyo, Japan. In addition, a comparative experiment was conducted using loess powder treated at 850 °C for 2 h and at 1050 °C for 1 h.

### 2.2. Infrared Absorption Spectra

Since raw loess contains useful minerals, infrared (IR) absorption spectra were investigated to confirm how these components change when loess is heated to a high temperature. The IR absorption spectra were measured for raw loess powder and heat-treated loess powder at 850 °C for 2 h and 1050 °C for 1 h. This experiment was performed using an FT-IR spectrometer at a test laboratory in Makino, Tokyo, Japan.

### 2.3. Low-Temperature Wet-Drying Method

Figure 1 shows the manufacturing process of functional loess balls using the low-temperature wet-drying method. In this study, a novel manufacturing method was attempted to utilize the useful properties of minerals and microbial enzymes contained in raw loess. Rather than following the conventional method of heat-treating loess at high temperatures, loess balls were produced by wet-drying methods at a low temperature of less than 90 °C. The resulting products were named loess bio-balls because they contain the moisture and useful minerals in contained in the original loess.

### 2.4. Radiant Intensity and Emissivity of FIR Emitted from Loess Bio-Balls

The radiant intensity and emissivity of the FIR emitted from loess bio-balls were measured at 40 °C as a function of 5–20 μm using an FT-IR spectrometer. This experiment was performed at the Korea Institute of Far-Infrared Ray Application Evaluation, Busan, Republic of Korea. In addition, the radiant intensity of the FIR emitted by the loess bio-balls between 20 °C and 60 °C was calculated using the radiant intensity of the loess bio-ball at 40 °C and the Stefan–Boltzmann law in Equation (1).

According to the Stefan–Boltzmann law, all matter vibrates at a given temperature and emits electromagnetic waves corresponding to the frequency of this vibrational motion [17].
(1)$$E\left(T\right)=\sigma eAT^{4}\left[W/m^{2}\cdot m\right]$$ where $\sigma =5.67\times 10^{-8}\;J/s\cdot m^{2}\cdot K^{4}$ is the Stefan–Boltzmann constant, e is the emissivity of the object, A is the area of the object, and *T* is the temperature of the object expressed as absolute temperature.

When heat is applied to the loess ball, the peak wavelength at which FIR is emitted maximally between 20 °C and 60 °C is obtained using Wien’s displacement law in Equation (2). The peak wavelength of the electromagnetic wave emitted by the loess ball at absolute temperature *T* is given by [17].
(2)$$\lambda_{max}=\frac{0.0029}{T}[m]$$

### 2.5. Transmitted Energy of FIR Passing through Five Materials

An FIR transmission experiment was performed to determine whether the FIR emitted from the loess bio-ball could pass through the various materials that can be used to manufacture the loess bio-ball mattress. These included wool cloth, cotton cloth, polyethylene terephthalate (PET), vinyl, and PET fillet. All different materials were cut to a thickness of 8 mm and placed in front of a loess bio-ball, and the transmitted energy and transmittance of FIR emitted from loess bio-ball through each material were measured. These measurements can provide information about whether FIR emitted from a loess bio-balls mattress can pass through the epidermis and subcutaneous tissue of the human body and reach the blood vessels, muscles, and organs beneath. The transmitted energy and transmittance of FIR passing through the materials were measured at 40 °C and in the 5.6–14 μm wavelength range using FT-IR spectrometer at the Korea Construction Material Test Institute (Test Method: KFIA-FI-1005), Seoul, Republic of Korea.

### 2.6. Blood-Flow Measurement Using Laser Doppler Flowmetry

Peripheral vascular tissue is sensitive to physiological changes in the body. The exact diameter of the arteriolar lumen is determined by neural and chemical regulation, and vasoconstriction and vasodilation of the arterioles are main mechanisms of blood-flow distribution to the capillary bed as well as regulation of systemic blood pressure. Hyperemia is the process by which the body adjusts blood flow to meet the metabolic needs of various tissues in health and disease. Careful control of the microcirculation in arterioles, capillaries, and venules is essential for life [18].

Laser Doppler flowmetry (LDF) is a continuous, non-invasive method of measuring blood flow in tissues using the Doppler shift of laser light as an information carrier [16]. Light (780 nm) from a laser diode is irradiated to the subcutaneous tissue through an optical fiber. The laser light is scattered and reflected by tissues and red blood cells flowing through the capillaries. The blood flow in the capillary network has a forward distribution of less than 1 mm/s or less. The principle of laser blood-flow measurement in capillaries is shown in Figure 2. The light reflected from the tissue is transmitted to the photodiode through another optical fiber and converted into an electrical signal I(t). The Doppler effect causes a change in the frequency of the received reflected light, which is proportional to the speed of the red blood cells. The frequency difference between light reflected from tissue (without Doppler shift) and light reflected from red blood cells flowing through capillaries (with Doppler shift) ranges from hundreds of hertz to tens of kilohertz. Therefore, two waves with approximately the same amplitude but slightly different frequencies overlap each other (causing reinforcing and destructive interference), creating a detectable beat signal.

Beat signal I(t) is produced by the superposition of two waves (with/without Doppler shift) of slightly different frequencies but identical amplitudes. The waves alternate in time between constructive and destructive interference, giving the resultant wave a time-varying amplitude enveloped with a beat frequency f_b_ and the wave itself with a frequency f_av_. By converting the beat signal I(t) from the time domain to the frequency domain using Fast Fourier Transform (FFT), the power spectrum density for frequency and the root–mean–square of time can be obtained. In the power spectrum graph of the beat signals [P(ω)], the frequency axis corresponds to blood-cell velocity and the power axis to the number of blood cells. The blood flow can be expressed as the total of the products of the number of blood cells traveling at each particular velocity multiplied by that velocity [19]. Therefore, blood flow can be calculated by dividing the integral of the product of the power spectrum and frequency of the beat signal by the root–square–mean of time as follows [20,21].
(3)$$Blood\;flow\approx \frac{1}{t_{rms}}\int \omega P\left(\omega \right)d\omega$$

### 2.7. Blood-Flow Measurement on Loess Bio-Ball Mat

The blood flow was measured at the left and right middle fingers (LMF, RMF) while raising the temperature of the loess bio-ball mat from 25 °C to 50 °C at 2.5 °C intervals. The experimental protocol was approved by the Inje University Bioethics Committee (registration number: INJE 2023-05-035-005) under the clinical trial registration entitled “Improvement of blood circulation by FIR emitted from loess bio-balls and health promotion effects in related diseases”. This experiment was mainly conducted at the R&D center of Hanwool Bio and also included self-reported home measurements from study participants.

#### 2.7.1. Participants

This study is an exploratory study conducted prior to clinical research. The recruitment of subjects for clinical research on blood circulation was published as an article in the local newspaper “Yangsan News Park” on 7 November 2023. The eligibility criteria for participation were those who were 30 years or older and who regularly experienced discomfort due to cold hands and feet or blood-flow disorders. The average/mean age and physical conditions of 40 participants (19 females, 21 males) were as follows: the age was 60.45 (±9.05) years, the height was 169.04 (±6.57) cm, the weight (mass) was 61.40 (±9.91) kg, and the BMI was 21.48 (±3.46) $kg/m^{2}$. All were nonsmokers, without any medical history of hypertension, asthma, or diabetes. None of the participants were on any medications that may affect the autonomic nervous system (ANS) function.

#### 2.7.2. Study Design

The presence and transformation of useful trace elements contained in the raw loess and heat-treated loess were confirmed using X-ray diffraction and IR absorption investigations. Through physical and chemical research on loess, loess balls are manufactured that can be widely used in daily life and for health-promotion purposes while preserving the original functions of the raw loess [22]. The spectral intensity and transmittance of FIR emitted from the manufactured red clay balls were obtained. In order to predict whether FIR can pass through the human body, the penetration intensity of FIR for five substances was conformed. Then, loess bio-ball mat was made using the manufactured loess balls, and the blood flow and epidermal temperature were measured in the middle fingers of both hands for 40 participants using laser Doppler flowmeter. Loess bio-ball mat with a radiant intensity of $3.74\times 10^{2}\;W/m^{2}$ around 9.5–9.8 μm was used as a FIR source. Blood flow was measured using a Doppler laser flowmeter (ALF21; Advanced Co. Ltd., Tokyo, Japan). The red diode laser (780 nm, 2 mW) applied to the reflective electrode can penetrate approximately 0.5–1 mm beneath the skin, and the electrode is attached to the middle fingers (LMF, RMF) using a bandage. Epidermal temperature was measured using an infrared thermometer (MM-GP100, Harbin Xiande Technology Development Co., Ltd., Shenzhen, China).

#### 2.7.3. Statistical Analyses

All statistical analyses for the control group (*n* = 30) and the experimental group (*n* = 30) were performed using IBM-SPSS Statistics 29.0.2.0 (demo version; IBM Corp., SPSS Inc., Armonk, NY, USA). The correlation coefficient (*r*) and *p*-value were calculated for blood-flow and epidermal temperature measured in LMF and RMF at each set temperature. A *p*-value < 0.01 was considered statistically significant. Data processing, graphs, and logistic fitting were prepared using Excel 2016 (Microsoft, Seattle, WA, USA).

## 3. Results and Discussion

### 3.1. X-ray Diffraction Patterns

Analysis of loess powder using X-ray Diffraction (XRD) confirmed that the peaks correspond to kaolinite (Al_2_SiO_2_O_5_(OH)_4_) and quartz (${SiO}_{2}$). Table 1 shows the percentage composition of minerals in loess powder. The main constituent minerals of loess were feldspar and quartz, and the clay materials were sericite and kaolin.

Figure 3 shows the XRD patterns of untreated loess powder and loess powder treated at two high temperatures. Untreated represents the loess powder containing raw components because it had not been heat-treated. The measurements of 850 °C 2 h and 1050 °C 1 h indicate the loess powders that have been heat treated at 850 °C for 2 h and 1050 °C for 1 h, respectively. The XRD pattern showed that the supernatant powder is halloysite (Al_2_SiO_2_(OH)_4_) and the precipitated powder is quartz (${SiO}_{2}$). Halloysite peaks appeared in the untreated sample but disappeared in the loess powder that was heat treated at 850 °C for 2 h and 1050 °C for 1 h. Kaolinites and halloysites have the same chemical composition but halloysite has a higher water content and is an efficient adsorbent of cations and anions. Due to its structure, halloysite can be used as filler in either natural or modified forms in nanocomposites. On the other hand, mullite ($3{Al}_{2}O_{3}\cdot 2{SiO}_{2}$) peaks appeared in the loess powder treated at 1050 °C for 1 h. Mullite is an aluminum silicate compound composed of aluminum oxide (${Al}_{2}O_{3}$) and silicon dioxide (${SiO}_{2}$). It is an alumina ceramic that is strong at room temperature and has a high heat resistance. Figure 3 shows that when loess powder is heated at a high temperature, useful minerals can be lost or converted into other elements.

### 3.2. Infrared Absorption Spectra

Figure 4 shows the IR absorption spectra of the loess powders according to the wave number and the wavelength. Two important peaks were observed around 3600–3700 ${cm}^{-1}\;$(2.6–2.7 μm) and 1005–1105 ${cm}^{-1}\;$(9.5–9.8 μm) in untreated loess powder. The small absorption peaks due to OH-stretching near 2.6–2.7 μm correspond to nitric oxide (NO), which serves as a signal molecule in the cardiovascular system and dilates blood vessels [23,24]. But the peaks due to OH-stretching disappeared in the loess powder heated at 850 °C for 2 h and 1050 °C for 1 h. In addition, large absorption peaks were detected around 9.5–9.8 μm in untreated loess powder. These are mainly due to Si-O vibrational motions (deformation, stretching, and bending) which can selectively absorb externally applied far-infrared rays with similar frequency. FIR emitted from loess is likely produced by these vibrational motions of Si-O and has a similar frequency to that of water molecules in the body. Therefore, FIR from loess could activate the vibrational motions of water molecules in the body, increase core temperature, and further activate the functions of cells and tissues [25]. However, these absorption peaks related to resonance vibrations of water molecules were significantly reduced in the loess powder that was treated at high temperatures. The +PC15wt.% represents the IR absorption spectrum of loess granule made by adding 15wt% of adhesive Portland cement to the loess powder. The IR absorption spectrum of +PC15wt.% closely resembled the IR absorption spectrum of unheated loess powder but had an additional absorption (indicated by red circle in Figure 4) from the adhesive material. The results of IR absorption spectra showed that untreated loess powder retains the functions of useful minerals and microbial enzymes in the original loess, but these are significantly lost when loess powder is treated at high temperatures. Therefore, the loess bio-ball manufactured by the low-temperature wet-drying method (hereafter this will be referred to as the LW method) would have better moisture absorption and deodorization properties than the loess ball made by the conventional baking method.

### 3.3. Low-Temperature Wet-Drying Method

The halloysite state in the loess powder (described in Figure 3 and Figure 4) has a similar structural and chemical composition to kaolinite, but it also contains water molecules between layers. In loess powder in Figure 4, small IR absorption peaks from OH stretching appeared at 2.7–2.7 μm, and large IR absorption peaks from Si-O vibrational motion were observed at 9.5–9.8 μm. Preliminary experiments confirmed that when loess is heated at a high temperature, the useful properties of loess are lost. Therefore, for practical application of loess in real life, it is necessary to implement a LW method in which loess soil is aged at a low temperature (less than 90 °C) for 6 months instead of baked at high heat. In order to differentiate it from the existing loess baked at high temperatures, loess balls manufactured using the WA method are termed loess bio-balls (or Jangsoo bio-balls), and these retain the useful minerals and microbial enzymes found in raw loess.

Figure 5 shows porous loess bio-balls produced by the LW method. These loess bio-balls have a relatively hard exterior shell but remain porous on the inside. In addition, the upper and lower surfaces of the loess bio-ball have holes that allow the passage of moisture, which contributes to the humidity-control function. When the external environment is dry, the moisture inside the loess bio-ball can escape to the outside, and vice versa; when the external environment is humid, moisture can flow into the loess ball [22].

### 3.4. Radiant Intensity and Emissivity of FIR Emitted from Loess Bio-Balls

Figure 6a shows the radiant intensity of the loess bio-ball at 40 °C (313 K) at a wavelength of 5–20 μm. The radiant intensity of the loess bio-ball was $3.74\times 10^{2}\;W/m^{2}$ at around 9.5–9.8 μm. A large absorption peak appeared near 9.5–9.8 μm, which was also previously observed in Figure 4 as a large absorption of FIR due to Si-O stretching. The radiant intensity emitted from the loess bio-ball overlaps with parts of the wavelength band known as growth rays (5.6~14 μm wavelength), which is essential for the survival of living organisms in nature [26]. The wavelength band emitted from the loess bio-ball also closely matches the vibrational movement of water molecules inside human body at room temperature [8,10].

Therefore, FIR radiation can be selectively absorbed as thermal energy by water molecules in human cells and tissues by the resonance of waves, and the rest can be transmitted to surroundings as vibration energy [11]. Figure 6b shows the emissivity of FIR emitted from the loess bio-ball compared to a black body as a function of wavelength. The average emissivity of FIR emitted from loess bio-ball was 0.927 (92.7%) in the 5–20 μm wavelength range. In comparison, human skin which is known to be an excellent absorber and emitter of FIR radiation, has an emissivity of 0.98 ± 0.01 for a wavelength range of 2–14 μm [27,28].

For the practical application of FIR emitted from loess bio-balls in real life, it is necessary to obtain the radiant energy of FIR in a wider temperature range. The radiant energy of FIR emitted from the loess bio-ball at temperatures ranging from 20 °C (293 K) to 60 °C (333 K) with 5 °C increments was calculated using the radiant energy at 40 °C and the Stefan–Boltzmann law (Equation (1)). Figure 7 shows the radiant energy and peak wavelength of FIR from the loess bio-ball as a function of temperature. The amount of radiated energy is proportional to the fourth power of absolute temperature (*T*). The radiant energy (expressed as black circles) may appear to increase gradually because it is obtained in a narrow temperature range, but in fact it is proportional to the fourth power of the absolute temperature. The radiant energy was calculated to be $2.87\times 10^{2}\;W/m^{2}\cdot \mu m$ at 20 °C and $4.79\times 10^{2\;}W/m^{2}\cdot \mu m$ at 60 °C, which represents a 165.74% increase. The radiant energy was $3.28\times 10^{2}\;W/m^{2}\cdot \mu m$ at 30 °C and $3.74\times 10^{2}\;W/m^{2}\cdot \mu m$ at 40 °C. Between 30 °C and 40 °C, which includes the physiological human body temperature, the radiant energy increased by 114.02%. Figure 7 also shows the peak wavelength (expressed as white squares) of FIR emitted from the loess bio-ball calculated at 5 °C interval from 20 °C (293 K) to 60 °C (333 K) using the peak wavelength at 40 °C and the Wien’s displacement law (Equation (2)). The peak wavelength of FIR emitted from the loess bio-ball is inversely proportional to the absolute temperature (*T*). The peak wavelength was 9.571 μm at 30 °C and 9.265 μm at 40 °C, close to the physiological human body temperature. The peak wavelength at a room temperature of 20 °C (293 K) was 9.891 μm, and the peak wavelength at a body temperature of 37 °C (310 K) was 9.348 μm. FIR in this wavelength range matches the vibrational energy of water molecules in the body [11] and is believed to increase the temperature of these water molecules and activate surrounding cells and tissues [8].

### 3.5. Transmitted Energy and Transmittance of FIR Emitted from Loess Bio-Ball for Five Materials

To confirm whether the FIR emitted from the loess bio-ball can penetrate through various material components of the loess bio-ball mattress as well as the epidermis, FIR penetration experiments were performed using five different materials with a thickness of 8 mm. Figure 8 shows the transmitted energy and transmittance of FIR for five materials. The original FIR energy and transmission from the loess bio-ball at 40 °C was $3.74\times 10^{2}\;$W/$m^{2}\cdot \mu m$ and 0.927, respectively. The transmitted energy of FIR was $3.63\times 10^{2}$, $3.63\times 10^{2}$, $3.62\times 10^{2}$, $3.58\times 10^{2}$, and $3.57\times 10^{2}\;W$/$m^{2}\cdot \mu m$ when it passed through wool cloth, cotton cloth, PET, vinyl, and PET powder, respectively. For comparison with values transmitted through five materials, the values ($3.74\times 10^{2}\;$W/$m^{2}\cdot \mu m$, 0.927) of the loess sample represent the radiant energy and emissivity of the FIR emitted from the loess ball. The y-axis on the right side of Figure 8 shows the transmittance of FIR emitted from the loess bio-ball through five materials. These results suggest that FIR emitted from the loess bio-balls can pass through the mattress and the skin to reach the underlying subcutaneous fat, muscle, blood vessels, and organs [8].

### 3.6. Energy Conversion between FIR Emitted from Loess Bio-Ball and Cellular Water Molecules, and Its Effect on Improving Blood Flow

In Figure 6a, the peak wavelength (9.5–9.8 μm) of FIR emitted from the loess bio-ball is due to Si-O motions (Si-O deformation, Si-O stretching, Si-O bending). The three vibrational modes of the water molecules and their fundamental frequencies (wavelengths) in liquid water at 25 °C are as follows: symmetric stretching (v1 = 3657 ${cm}^{-1},\;$2.734 μm), bending (v2 = $1595\;{cm}^{-1}$, 6.269 μm), and asymmetric stretching (v3 = $3756\;{cm}^{-1}$, 2.662 μm) [29,30]. The wavelength of FIR in this region (9.5–9.8 μm) matches the absorption wavelength due to the motions (the bending of v2 and the combination of v2 and libration) of cellular water molecules. On the other hand, the estimated FIR intensity of around 2.7–2.8 μm (this is below the measurement range) in Figure 6a is caused by water molecules (OH stretching as shown in Figure 4) contained in the halloysite (${Al}_{2}{Si}_{2}O_{4}{\left(OH\right)}_{4}$ of the untreated loess. FIR in this region is selectively and strongly absorbed by water molecules undergoing OH-stretching movements. FIR emitted from loess bio-balls is absorbed as higher vibrational energy by the selective absorption of water molecules in the body. As water molecules in cells and tissues vibrate more actively, the body temperature can be increased in the body’s core. An increase in body temperature can activate the thermoregulatory mechanism in the body and lead to the vasodilation of vessels, which, in turn, increases blood flow.

### 3.7. Blood Flow Measured at LMF and RMF When Using Loess Bio-Ball Mat

Figure 9 shows blood-flow (squares) and epidermal temperatures (circles) in LMF when using a loess bio-ball mat. When the set temperature applied to the mat was increased from 25 °C to 50 °C at 2.5 °C intervals, blood flow significantly increased by 39.01%, from 37.48 ± 6.49 mL/min to 52.11 ± 6.88 mL/min. In addition, epidermal temperature increased by 7.40% (2.53 °C), from 34.20 °C to 36.73 °C. The correlation coefficient *(r*) of blood flow according to the set temperature measured in LMF was 0.785–0.998 and the *p*-values were less than 0.01. The *r* of the epidermal temperature according to the set temperature measured in LMF was 0.391–0.979 and the *p*-values were less than 0.01.

Figure 10 shows blood-flow (squares) and epidermal temperatures (circles) at RMF when using a loess bio-ball mat. When the set temperature applied to the mat was increased from 25 °C to 50 °C at 2.5 °C intervals, blood flow significantly increased by 39.62%, from 37.15 ± 5.04 mL/min to 51.87 ± 6.40 mL/min. In addition, epidermal temperature increased by 7.46% (2.55 °C), from 34.20 °C to 36.75 °C. The *r* of blood flow according to the set temperature measured in RMF was 0.497–0.996 and the *p*-values were less than 0.01. The *r* of epidermal temperature according to the set temperature measured in RMF was 0.593–0.961 and the *p*-values were less than 0.01.

In this study, an experiment was conducted to detect changes in the physicochemical properties of raw loess powder and loess powder baked at high temperature. After analyzing the experimental results, a novel LW method of manufacturing loess bio-balls was proposed to preserve the useful minerals found in raw loess. These loess bio-balls can have practical application in real life for health-promoting purposes.

Body temperature is regulated by the hypothalamus sensing blood temperature around receptors in the arterioles [31,32]. The thermoregulatory center in the hypothalamus maintains a temperature of 37 °C (98.6 F), which is optimal for enzyme function [33,34]. The skin also contains temperature receptors that send feedback to the thermoregulatory center. When the body’s core temperature rises above 37 °C, the receptors send signals to the hypothalamus, which transmits nerve impulses to the skin arterioles, causing vasodilation and increased blood flow [35]. FIR with a wavelength of 9.5–9.8 μm emitted from the functional loess bio-balls causes the water molecules in the body to vibrate more actively, increasing the temperature of the core in the body. As shown in Figure 9 and Figure 10, when the core temperature rises, the blood-flow and epidermal temperature also increase accordingly to release excess heat.

Recent other studies on FIR have indirectly measured blood flow after applying FIR to specific body parts, such as the feet and hands. These studies mainly detected changes in the biomarkers in the body before and after FIR application [4,11,29]. In this study, changes in the blood-flow and epidermal temperature according to the set temperature were simultaneously measured while irradiating the entire body with FIR emitted from loess bio-balls. Our findings show that the effects of FIR on blood flow are effectively reflected in real-time rather than before or after FIR exposure. Changes in the blood flow and epidermal temperature were analyzed using the energy conversion between the FIR and water molecules and the thermoregulatory mechanism of the hypothalamus.

## 4. Study Limitations and Future Directions

It was confirmed that FIR emitted from functional bio-balls increases the core temperature in the body, thereby increasing blood circulation and, thus, the temperature of the epidermis. Although this study was conducted on healthy subjects, FIR from bio-balls is expected to be helpful in blood-circulation disorders such as cold hands and feet. If systematic clinical research can be targeted on patients with specific diseases in a clinical setting, the therapeutic effects of FIR emitted from loess bio-balls can be better established and applied in various fields including alternative and complementary medicine.

## 5. Conclusions

The peak wavelength of the FIR emitted from the loess bio-ball was 9.5 to 9.8 μm, which closely matches the wavelength of the vibrational motion of water molecules in the human body [8]. Therefore, FIR emitted from the loess bio-ball can penetrate the epidermis and adipose tissue, and transmit radiation energy to the water molecules inside the human body through the resonance effect. This can increase the core temperature and activate water molecules in cell membranes and tissues [25]. The FIR emitted from loess bio-balls can help improve blood circulation through the thermoregulatory action of the hypothalamus. The blood flow measured in the middle fingers of both hands in 40 participants increased by 39.01~39.62%, and the epidermal temperature increased by 2.53~2.55 °C.

The loess bio-balls manufactured in this study are materials that can be widely applied to daily life and various health-promotion products. FIR emitted from loess bio-balls can be easily applied in various forms for therapeutic purposes in medical settings. FIR therapy is already commonly used as a non-invasive treatment for diseases such as myocardial ischemia, diabetes, and chronic kidney disease [36]. In particular, the FIR (around 2.7–9.8 μm) emitted from loess bio-ball balls is the same frequency (wavelength) as that of the vibrating water molecules in the body. Therefore, it is predicted that this FIR will not only increase the temperature of the core of the body and further activate cells and tissues, but also stimulate lymph circulation and discharge waste from the body, thereby reducing inflammation levels.

## Figures and Tables

**Figure 1 bioengineering-11-00380-f001:**
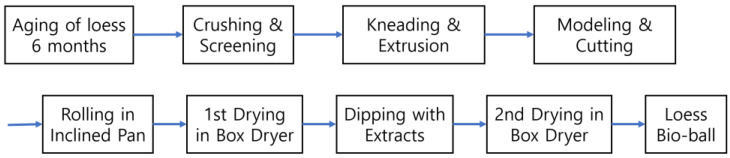
Flow chart outlining the manufacturing process of functional loess bio-balls.

**Figure 2 bioengineering-11-00380-f002:**
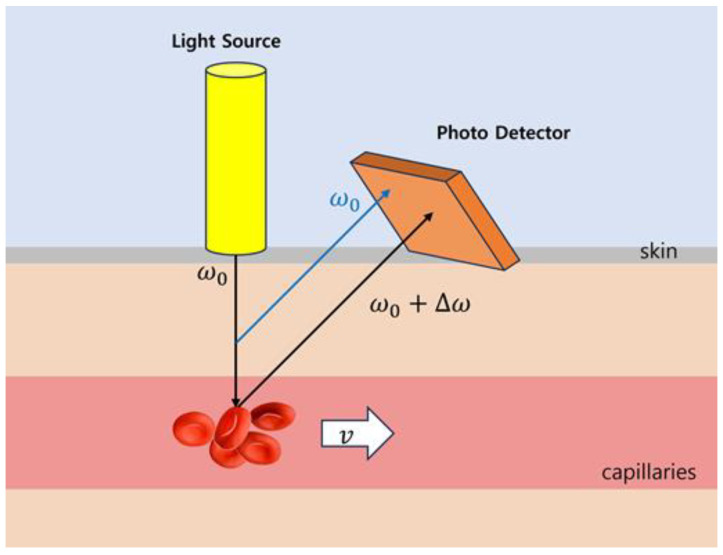
Principles of PDF. The frequency of light reflected by tissues in a stationary state does not change, but the light reflected by red blood cells flowing through capillaries changes frequency due to the Doppler effect.

**Figure 3 bioengineering-11-00380-f003:**
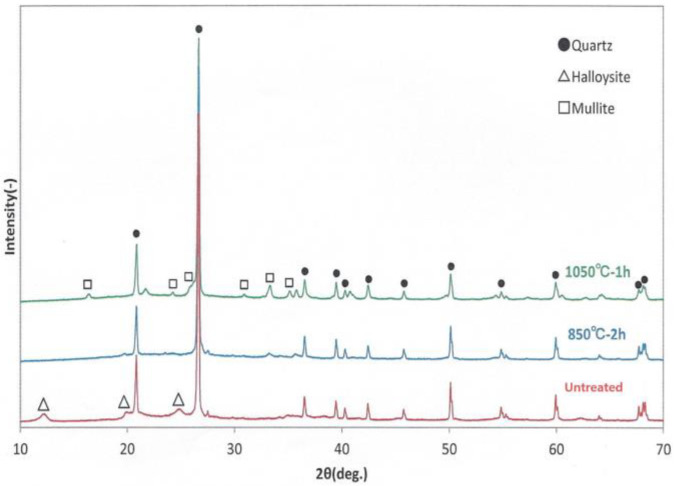
X-ray diffraction patterns of untreated loess powder and heat-treated loess powder.

**Figure 4 bioengineering-11-00380-f004:**
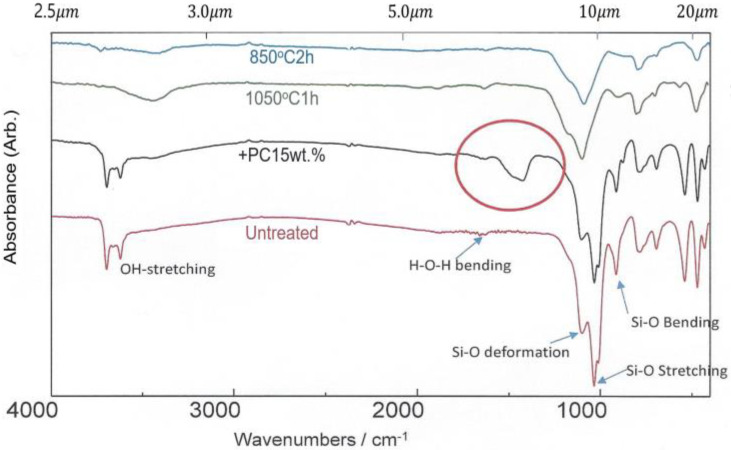
IR absorption spectra of loess powder at various temperatures.

**Figure 5 bioengineering-11-00380-f005:**
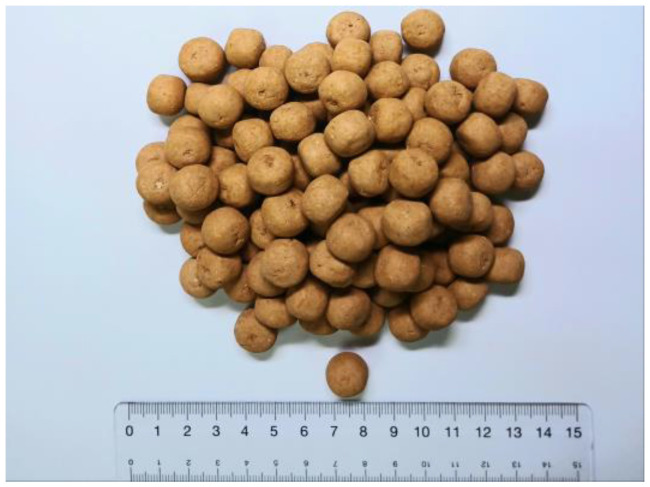
The loess bio-balls manufactured by the LW method. The diameter of the loess bio-ball is about 1.3 mm.

**Figure 6 bioengineering-11-00380-f006:**
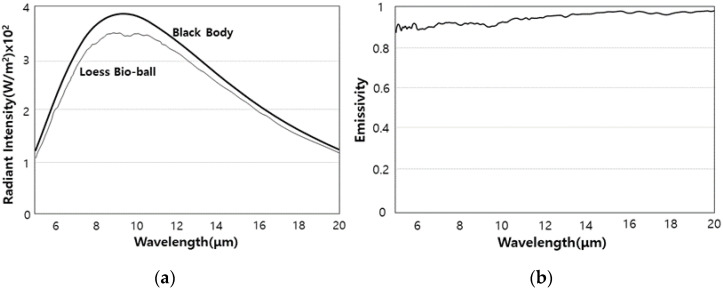
(**a**) Radiant intensity and (**b**) emissivity of FIR emitted from loess bio-ball at 40 °C (313 K) as a function of wavelength.

**Figure 7 bioengineering-11-00380-f007:**
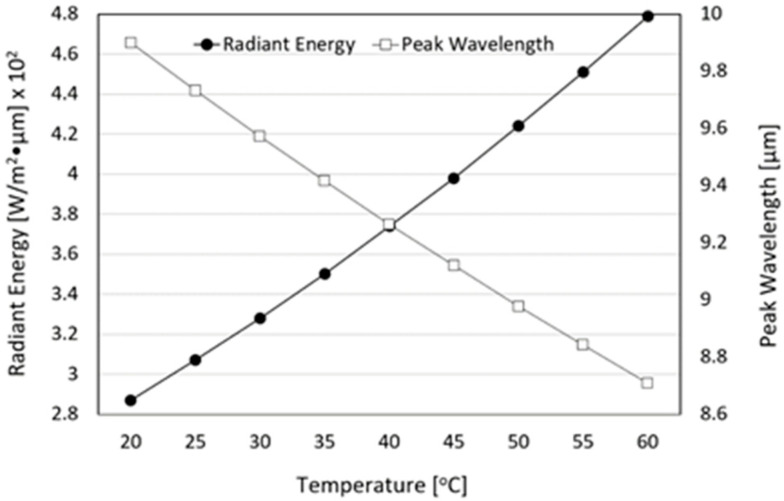
Radiant energy and peak wavelength of loess bio-ball as a function of temperature.

**Figure 8 bioengineering-11-00380-f008:**
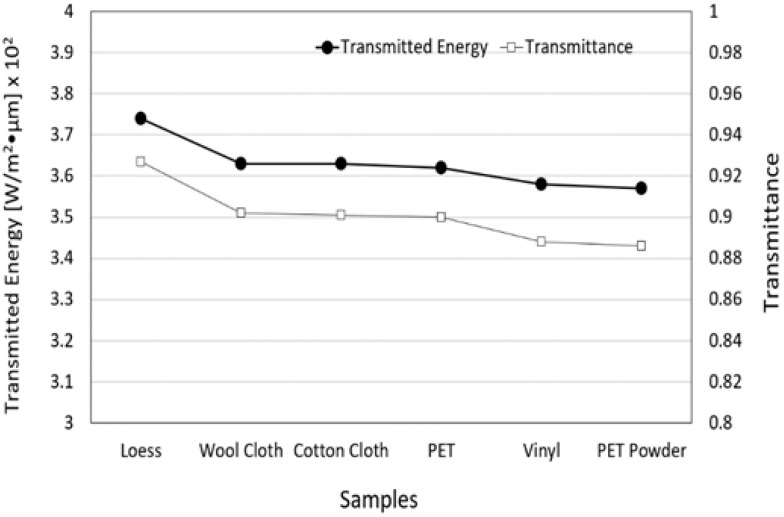
Transmitted energy and transmittance of FIR emitted from loess bio-ball for five materials. For comparison with values transmitted through materials, the values ($3.74\times 10^{2\;}$W/$m^{2}\cdot \mu m$, 0.927) of the loess sample represent the radiant energy and emissivity of the FIR emitted from loess bio-ball.

**Figure 9 bioengineering-11-00380-f009:**
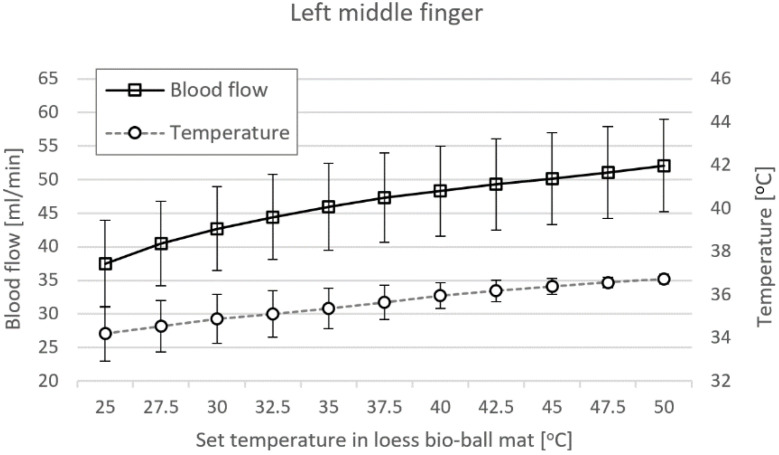
Blood flow and epidermal temperature at LMF when using loess bio-ball mat.

**Figure 10 bioengineering-11-00380-f010:**
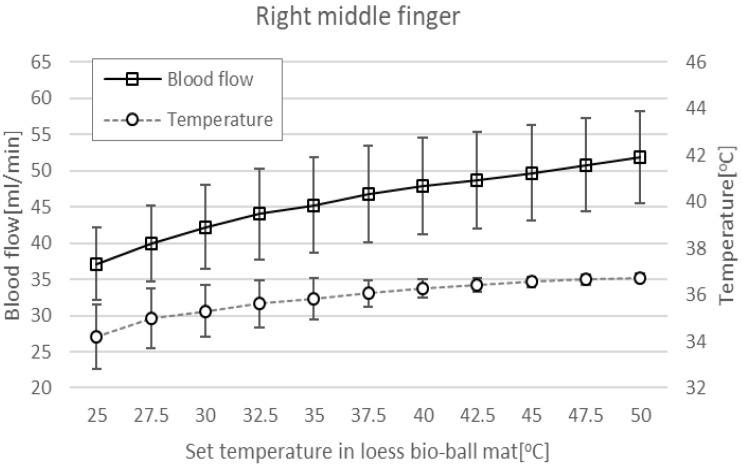
Blood flow and epidermal temperature at RBT when using a loess bio-ball mat.

**Table 1 bioengineering-11-00380-t001:** Chemical composition (%) of loess powder obtained by XRD measurement.

Analyte	Result	Line	Net Int.	BG Int.
SiO_2_	72.3262%	SiKa	902.218	2.766
Al_2_O_3_	15.9731%	AlKa	280.657	12.753
Fe_2_O_3_	5.9951%	FeKa	349.073	1.330
K_2_O	2.1878%	K Ka	102.760	0.742
MgO	1.0557%	MgKa	6.590	0.480
TiO_2_	0.9764%	TiKa	13.524	0.169
CaO	0.4570%	CaKa	16.625	0.434
P_2_O_5_	0.3802%	P Ka	7.140	0.619
MnO	0.1914%	MnKa	8.758	0.787
Na_2_O	0.1912%	P Ka	0.608	0.113

Int. is an abbreviation for intensity, and BG represents background intensity.

## Data Availability

The data supporting the results of this study can be provided by the corresponding author upon reasonable request.

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
