# Peer review of "Characteristics of Far-Infrared Ray Emitted from Functional Loess Bio-Balls and Its Effect on Improving Blood Flow"

_bioengineering, 2024, doi:10.3390/bioengineering11040380_

Round 1

Reviewer 1 Report

Comments and Suggestions for Authors

The article is very interesting and it could be accepted for publication after very minor revision 

- Include in the abstract some numerical results 

-Include in the abstract tow sentence stating the conclusion and recommendation(s) releated to the findings.

- Enhance the resolution of both Figures 3 and 4.

-Add scale bare for Figure 5

Author Response

  1. In the abstract, blood flow and epidermal temperature measured in the middle fingers of the left and right hands at 20°C and 50 °C in 40 participants using the loess bio-ball mat are presented.
  2. These findings can be applied to the improvement of diseases related to blood flow, such as cold hands and feet and diabetes of the feet.

  3. Figure 3 and Figure 4 have been replaced with the original files with improved resolution.

  4. In Figure 5, a ruler with a scale is shown to easily determine the size of the loess bio-ball.

  5. In addition, the co-author(J.H.Kim), a image processing expert, modified the figures, and the co-author(M.S. Kim), a native English-speaker (MD), corrected the manuscript.

Reviewer 2 Report

Comments and Suggestions for Authors

This manuscript presents a study on the properties of loess, a sediment known for its health-promoting effects, and its ability to emit far-infrared rays (FIR) when manufactured into bio-balls. The study investigates the XRD and IR of raw and heat-treated loess, proposing a novel manufacturing technique involving low-temperature and wet-drying methods to preserve loess's beneficial properties. Experiments on the radiant energy and transmittance of FIR demonstrate that FIR emitted from loess bio-balls can enhance blood flow by being selectively absorbed by water molecules, raising the body's core temperature. An exploratory study with 40 participants showed significant increases in blood flow and epidermal temperature upon exposure to FIR from loess bio-balls.

Publication is recommended after addressing the following questions and comments:

  1. Including a control group or comparing the effects of loess bio-balls with other materials emitting FIR at different intensities or wavelengths could strengthen the argument about the uniqueness and efficacy of loess bio-balls. Have other materials or control conditions been considered to highlight the specific advantages of loess?
  2. Further clarification on the underlying biological mechanisms, possibly through molecular or cellular studies, could enhance understanding. How do these interactions at the molecular level lead to observed physiological changes?
  3. Information on the long-term effects of regular exposure to FIR from loess bio-balls, as well as the repeatability of results over time, would be valuable for assessing their potential for chronic conditions or long-term therapy.
  4. Further details on the statistical methods, including any assumptions made, model validations, and the handling of potential outliers or variations in participant responses, would strengthen the credibility of the results.
  5. Positioning loess bio-balls in the context of existing therapies or products that aim to improve blood flow or utilize FIR would help highlight their unique value proposition. Are there advantages in terms of efficacy, cost, or safety compared to current market options?

Author Response

1. In a comparative experiment using a conductive electric mat, when the set temperature was increased from 25℃ to 49℃, there was little change (0.91-1.02% increase) in blood flow measured in the middle fingers of the left and right hands, and the change in epidermal temperature was minimal (1.35% increase). Since this study deals with the characteristics of functional loess bio-balls manufactured using a low-temperature wet-drying method and the increase in blood flow and epidermal temperature, the measurements obtained from comparative experiments (conductive electric mat) with 40 participants were excluded from this paper. For the comparison group, blood flow and epidermal temperature measurements using a carbon mat that emits FIR are considered to be sufficiently valuable and will be covered in follow-up experiments.

2. In this study, the comparison of the physical properties of loess bio-balls and the effect of improving blood flow were focused on. We believe that research on biological mechanisms at the molecular or cellular level that you pointed out is valuable. A preliminary study was conducted on the changes in bio-impedance (intracellular fluid (ICF), extracellular fluid (ECF), cell capacitance (Cm), phase angle (PA), etc.) before and after using the loess bio-ball mat. These studies will be conducted in detail in subsequent experiments.

3. According to experiments performed more than 200 times, blood flow responds to far-infrared irradiation in real time, while lymphatic fluid (waste disposal) moves very slowly in lymphatic vessels (1cm/min.). Therefore, the lymphatic effect appears only after exposure to FIR for a long time. For example, when the loess bio-ball mat was used at 40℃ for 30-40 minutes, there was little change in impedance parameters, whereas when used at 28-30℃ for 7 hours, the ICF/ECF ratio increased by 3.8-4% due to decreased ECF (inflammatory fluid). Precise comparative measurements and analysis using more impedance parameters will be supplemented and published in an academic journal in the future.

4. The statistical method you pointed out is described in the paper. In addition, the validity of the results of this study was proven by using the IBM-SPSS Statistics 29.0.2.0 as a result evaluation model. Research model is applied as follows: FIR (9.5 - 9.8μm) emitted from a loess bio- ball increase the core temperature of the body as water molecules selectively absorb the energy of FIR. When the core temperature exceeds 37°C, the blood flowing into the capillaries increases due to the thermoregulatory mechanism of the hypothalamus, which will lead to an improvement in blood flow and epidermal temperature.

5. Existing products (electric mats, heated mats, stone beds, etc.) mainly deal with the effect of increasing the surface temperature due to conductive heat. In this study, the results of using loess bio-balls showed that the effect of improving blood flow and increasing epidermal temperature was shown due to the increase in core temperature caused by far-infrared rays (9.5-9.8μm).

While some FIR lamps are used as devices to improve blood flow in existing medical settings, little study is being conducted on loess bio-ball mats. In addition, research on the increase in surface temperature has been conducted using some thermal imaging cameras. However, there has been little research on raising the core temperature with far-infrared rays and improving blood flow and waste through the body's temperature regulation mechanism.

The loess bio-ball mat is more expensive than existing low-cost/conductive mats. However, it can be applied to health promotion in various forms because it improves blood flow and increases epidermal temperature. This has a heating element beneath the loess balls and is shielded with a safety covering fabric/device. Therefore, even if the temperature applied to the mat is set to 50°C, the actual temperature of the cushion placed on the mat is around 37°C, ensuring user safety.

Reviewer 3 Report

Comments and Suggestions for Authors

The article is new and interesting. The relevance is beyond doubt. The development of analytical methods for studying materials and their applications is a promising task. In terms of volume and subject matter, this work meets the requirements of the Journal. There are some questions for improvement: 1. It is necessary to more clearly formulate the practical significance of the results obtained. 2. A more detailed comparison with literary sources is desirable. 3. It is advisable to make conclusions more concise. 4. It is advisable to consider in more detail the application of the results obtained.

Author Response

1. In this study, the effects of improving blood flow and epidermal temperature were compared and reviewed using the conventional conductive mats (electric mat, water heated mat, stone bed) and the loess bio- ball mat. However, the electric mat and heated mat did not have a smooth temperature control function. Although the conductive electric mat slightly increased the epidermal temperature (1.35%), it had little effect on improving blood flow (0.91-1.02%). Stone beds can also slightly increase epidermal temperature, but have no effect on improving blood flow.

This paper deals with the characteristics and blood flow effects of functional loess bio-balls manufactured using a low-temperature wet-drying method to maintain the beneficial therapeutic functions of raw loess. Therefore, This paper describes the effect of improving blood (also epidermal temperature) when using the loess bio-ball mat.

2. A detailed comparison with literature data related to various diseases related to blood flow has been added in the manuscript.

3. The abstract was changed to a structured abstract. 3. Results and 4. Discussion were integrated into 3. Results and Discussion (reduced). 5. Conclusion was organized. 

4. It is believed that the effect of improving blood flow can be applied to various diseases, such as cold hands and feet, diabetes, lymphatic circulation disorders, menstrual pain, and chronic pain diseases. Further research on these diseases is scheduled to be promoted in the near future.

5. In addition, the co-author, an image processing expert, modified the figures, and the co-author, a native English-speaking doctor, corrected the manuscript.